# Are Convolutional Networks Inherently Foveated?

**Bilal Alsallakh**     **Vivek Miglani**     **Narine Kokhlikyan**

**David Adkins**     **Orion Reblitz-Richardson**

Meta AI

## Abstract

When convolutional layers apply no padding, central pixels have more ways to contribute to the convolution than peripheral pixels. Such discrepancy grows exponentially with the number of layers, leading to implicit foveation of the input pixels. We show that this discrepancy can persist even when padding is applied. In particular, with the commonly-used zero-padding, foveation effects are significantly reduced but not eliminated. We explore how different aspects of convolution arithmetic impact the extent and magnitude of these effects, and elaborate on which alternative padding techniques can mitigate it. Finally, we compare our findings with foveation in human vision, suggesting that both effects possibly have similar nature and implications.

## 1   Background and Motivation

Tsotsos et al [20] described the boundary problem in visual processing pyramids as follows: "central items will appear to be stronger at the output layer than items in the visual periphery even if the peripheral items are in fact stronger. This is due solely to the numbers of connections feeding units in successively higher layers in the outer regions of the pyramid". Two main approaches have been considered in the artificial-intelligence literature to address the boundary problem: attention mechanisms akin to saccadic eye movements [11, 17], and extending the input to each layer in the pyramid by means of padding as in classical image processing [2, 16].

Since the inception of AlexNet [10], the majority of Convolutional Neural Networks (CNNs) have used padding as a convenient way to address the boundary problem. The padding amount is selected so that the output of the convolutional layer is equal to its input. This padding scheme is often called `SAME` padding. Maintaining the size of feature maps computed over multiple convolutional layers is an essential property in various architectures such as ResNet [8] where controlled reduction of this size is performed at pooling or strided-convolution layers.

In their analysis of the effective receptive field (ERF) of CNNs, Luo et al [13] noted that it has a Gaussian shape and suggested that CNNs "may automatically create a form of foveal representation". In this work, we examine the above-mentioned effects more closely, focusing on how different aspects of convolution arithmetic [6] impact such foveation effects in CNNs. We demonstrate how:

- `SAME` zero padding incurs a mild foveation in CNNs, compared with no padding. The extent and magnitude of these effects are governed by various aspects of convolution arithmetic.

- Circular padding is the only `SAME` padding method that treats all pixels uniformly. Mirror padding warrants equal connectivity for all pixel, however, through different pathways.

We finally discuss the similarities of these effects with foveation in human vision.

3rd Workshop on Shared Visual Representations in Human and Machine Intelligence (SVRHM 2021) of the Neural Information Processing Systems (NeurIPS) conference, Virtual.

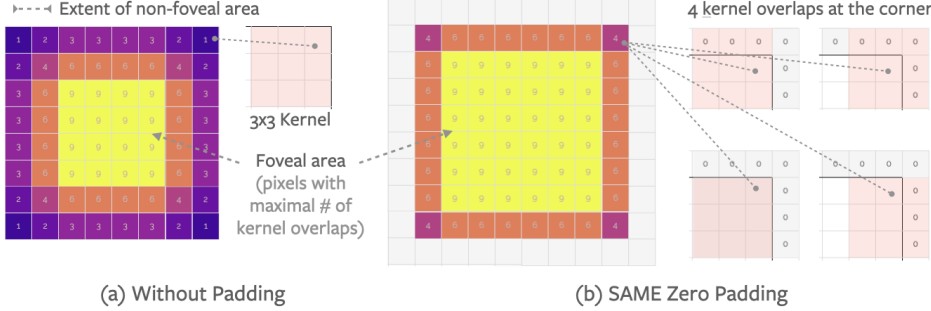

(a) Without Padding  (b) SAME Zero Padding

Figure 1: Foveation behavior when performing $3 \times 3$ convolution: (a) Without padding, the boundary pixels can only overlap a subset of all $3 \times 3$ kernel cells as the numbers indicate, unlike interior pixels. (b) This effect is alleviated under 0-padding: Corner pixels overlap the kernel 4 times instead of 1.

## 2 The Boundary Problem and Foveation

We refer to the marginalization of periphery due to the aforementioned boundary problem as foveation. Note that the pixel density is still uniform in the input space, unlike photoreceptors in human retina. Refer to Section 5 for discussion on the relation between the effects we expose in CNNs and foveated human vision.

Figure 1 illustrates how SAME zero padding alleviates foveation effects when performing $3 \times 3$ convolution without eliminating them. When no padding is applied (Figure 1a), these effects extend over a two-pixel region at the boundary. In contrast SAME zero padding limits the effects to the boundary pixels and increases the possibilities of these pixels to overlap the kernel as Figure 1b illustrates for one corner pixel. Figure 2 illustrates how the extent and magnitude of foveation effects increase over successive convolutional layers. We next explore in detail how different aspects of convolution arithmetic impact both quantities.

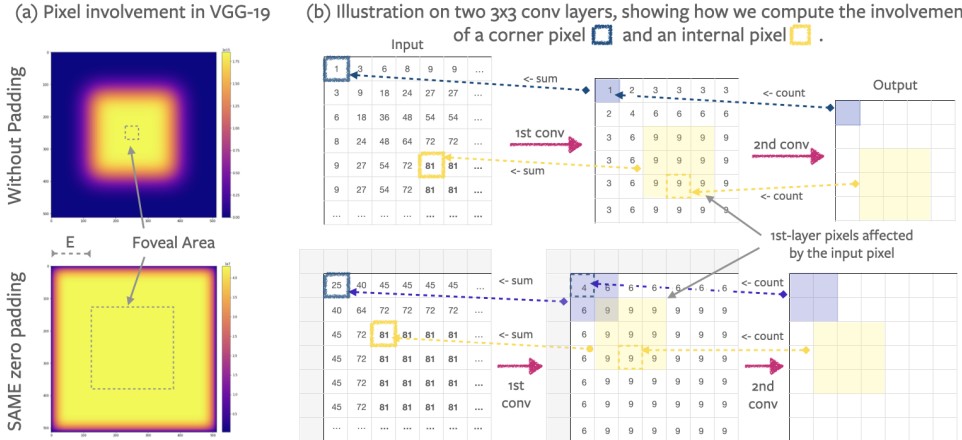

Figure 2: Foveation behavior over successive layers, with and without SAME zero padding. (a) Visualizing the effect for VGG-19 [19] with a $512 \times 512$ input space. Color represents the number of paths to the output layer for each input pixel. (b) Illustrating the effect on two convolutional layer for simplicity. The depicted counts indicate the number of paths to the output layer for each input pixel.

## 3 The Extent of Non-Foveal Area in CNNs

By foveal area we mean the set of input pixels having the maximal number of possible connections to the output layer for any given input size (Eq. 3). This number is equal to 9 in Figure 1 and to 81 in Figure 2b. We define the extent of non-foveal area as the pixel distance between the boundary and

the foveal area in a sufficiently large input space. By sufficiently large, we mean an input size for which the foveal area is nonzero. Two types of layers dictate this extent in CNNs:

- Standard convolutional layers with a stride of 1 increase the extent in an *additive* way, depending on their kernel size $k_i$ and the padding amount $p_i$.
- Downsampling layers such as pooling layers and strided-convolution layers increase the extent in a *multiplicative* way for layers that *succeed* them, depending on the stride $s_i$.

More formally, for a sequentially applied list of convolutional and downsampling layers $[L_1, ..., L_n]$ the extent of foveation, $E$, is computed as follows (where $d_i$ denotes the dilation factor at layer $L_i$):

$$E = \sum_{i=1}^{n} \big(d_i \cdot k_i - p_i - s_i - d_i + 1\big) \cdot \prod_{j=1}^{i-1} s_j \tag{1}$$

**Example:** ResNet-50 performs two downsampling operations, followed by four blocks $B_1, .., B_4$ of convolutional layers. Three additional downsampling operations occur between these blocks, all of which use a stride of 2. The additive term in Eq 1 is equal to 1 for all standard convolutional layers based on their parameters $k_i = 3$, $s_i = 1$, $d_i = 1$, and $p_i = 1$. Accordingly, the additive term for each block is equal to $|B_i|$, the number of these layers in block $B_i$. The extent of non-foveal area is hence equal to (we use a recursive version of Eq 1):

$$E_{\text{ResNet-50}} = 2 \times 2 \times \Big(|B_1| + 2 \times \big(|B_2| + 2 \times (|B_3| + 2 \times |B_4|)\big)\Big)$$

$$= 4 \times \Big(3 + 2 \times \big(3 + 2 \times (5 + 2 \times 2)\big)\Big) = 180 \tag{2}$$

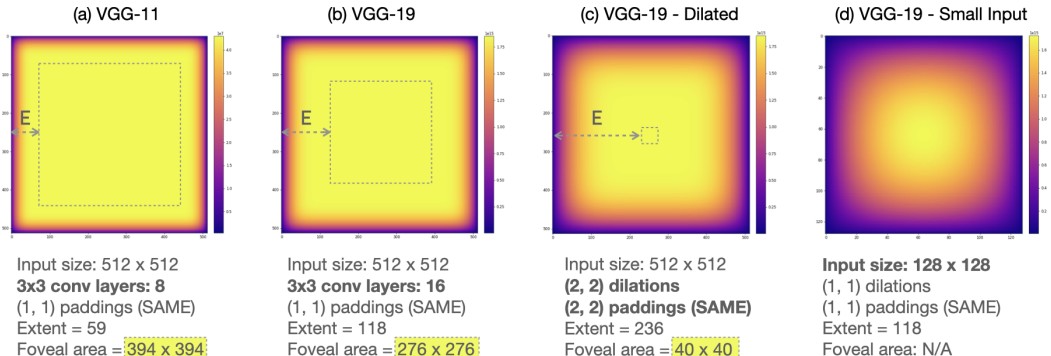

Figure 3: The foveation behavior of four VGGNet [19] variants. (a) A shallow VGG. (b) Doubling the number of layers. (c) Dilating all filters in VGG-19. (d) A $4 \times 4$ smaller input, compared with (b).

**Impact of kernel size and dilation:** A larger kernel increases the additive part in Eq 1. Likewise, a dilation factor larger than 1 effectively extends the kernel, and subsequently the non-foveal area. Figure 3c demonstrates this effect where we assign a dilation factor of 2 to all layers in VGG-19. This doubles the extent of the non-foveal area since the additive part evaluates to 2 instead of 1. Note that we increase the padding area to retain SAME padding.

**Impact of strides:** Downsampling layers amplify the extent of the non-foveal area incurred by *subsequent* convolutional layers in a multiplicative way. The amplification factor is equal to the stride which typically is equal to 2. Multiple pooling operations can hence rapidly increase the extent.

**Impact of input size:** The extent of non-foveal area $E$ is noticeably independent of input size, according to Eq 1. This means, the extent *relative* to input size is larger for smaller inputs as Figure 3d demonstrates. Note how the foveal area vanishes when the input size is smaller than $2E \times 2E$. Nevertheless, the input size can still impact the extent of foveation due to effects unaccounted for in Eq 1. Specifically, the input size was shown to determine the presence of uneven padding or uneven erosion of feature maps in CNNs [1]. For example, unlike a $257 \times 257$ input, a $256 \times 256$ input causes padding to be left out at the right and bottom sides in every downsampling layer in ResNet-50, as Figure 4a illustrates. This increases the extent of the non-foveal area at these two sides, compared with the left and top sides (Figure 4c), and shifts the center of foveation to the left and top.

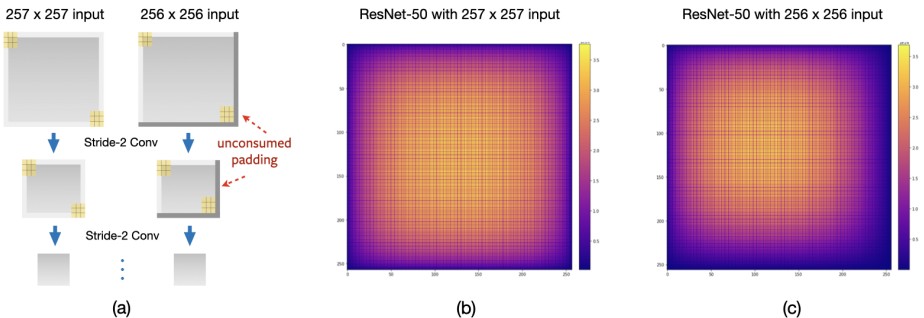

Figure 4: (a) Input size can lead to uneven treatment of the borders. When this occurs, the center of foveation is shifted as depicted in (c), unlike (b). The checkerboard patterns are caused by the $3 \times 3$ downsampling kernels in ResNet whose size is not divisible by the stride [15].

# 4 The Magnitude of Foveation in CNNs

The maximal number of connections an input pixel can have to the output layer is equal to:

$$C_{\max} = \prod_{i=1}^{n} \lceil \frac{k_i}{s_i} \rceil^2 \qquad (3)$$

Pixels that reach this number have a distance larger than $E$ from any border and can hence only appear in an input of size larger than $2E \times 2E$. Other pixels do not reach this number because the inverted sub-pyramid [20] that belongs to such a pixel $(x, y)$ crosses the boundary at an intermediate layer $L_i$. This sub-pyramid contains all pixels in all layers that are impacted by the input pixel. The earlier the layer $L_i$, the smaller the number of connections $C_{(x,y)}$. This happens as $(x, y)$ gets closer to an edge or to a corner.

Padding plays a fundamental role in alleviating the connectivity reduction for pixels close to the boundary. To appreciate this role, consider the number of connections the corner pixel $(1, 1)$ has to the output layer. When no padding is applied at any layer, the inverted sub-pyramid is aggressively pruned, leaving a single path to the output (i.e. $C_{(1,1)}^{\text{no\_pad}} = 1$) through the same corner in successive layers as the top row in Figure 2b illustrates. In contrast, under zero padding, this number grows rapidly over layers as the pruning leaves connections through $2 \times 2$ pixels in the first layer, through $3 \times 3$ pixels in the second layer, and so on. The total number of connections $C_{(1,1)}^{\text{padded}} << C_{\max}$ remain small compared with a pixel from the foveal area.

Figure 5a plots the number of connections $C_{i,i:1 \leq i \leq 512}$ through VGG-19 for diagonal pixels in a $512 \times 512$ input, both under zero padding and without padding. When no padding is applied, the plotted series expectedly resembles a binomial distribution, since the accumulation of counts in the inverted sub-pyramids is analogous to that in Pascal's triangle. With a large number of layers, the distribution becomes Gaussian, as was shown by Luo et al [13].

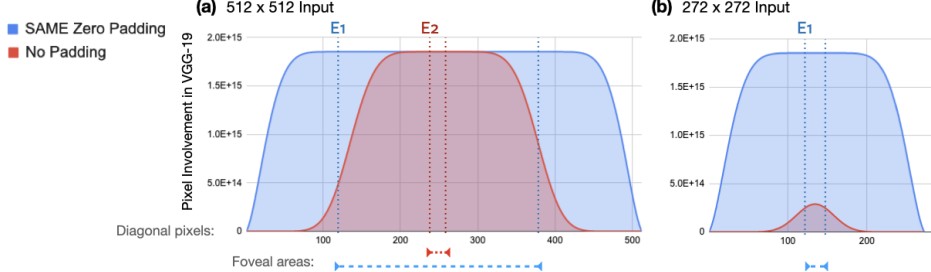

Figure 5: The magnitude of foveation in VGG-19. The plots show the involvement of different pixels along the diagonal under two padding modes. (a) An input larger $2E \times 2E$ under both padding modes. (b) An input smaller than $2E \times 2E$ when no padding is applied.

Figure 5b plots the same quantities as in Figure 5a, using a $272 \times 272$ input. This input is significantly smaller than $2E \times 2E$ when no padding is applied ($E = 236$). Compared with zero padding, the overall pixel involvement is drastically reduced. This explains the strong impact of accuracy observed in [9] on image classification and segmentation when no padding is applied.

It is worth mentioning that applying padding mitigates connectivity reduction, even when the convolution arithmetic yields the same foveal area for variants with and without padding. This is evident by comparing Figure 3c with the top plot in Figure 2a due to the aggressive pruning explained earlier in this section.

## 5 Discussion and Takeaways

We demonstrated that CNNs are inherently foveated even when `SAME` zero padding is applied at all convolutional layers. Three padding alternatives can eliminate or mitigate foveation effects in CNNs:

- **Circular padding** renders the input as infinite to the kernel. Once the kernel hits one side, it can seamlessly operate on the pixels of the opposite side (Figure 6a). This warrants that border pixels and interior pixels are treated equally, fundamentally eliminating foveation.
- `SYMMETRIC` **reflection padding** warrants each pixel an equal number of paths to the output layer. Figure 6b illustrates this fact for a corner pixel, where the reflection copies recover the number of kernel-pixel overlaps to 9 (refer to the appendix for detailed illustrations). However, unlike with interior pixels, these overlaps do not utilize the $3 \times 3$ kernel cells uniformly. The right and bottom cells are left unutilised.
- `FULL` **zero padding** uses twice the amount of `SAME` padding to warrant all possible overlaps between boundary pixels and the kernel. While this eliminates foveation, the output size is increased at each convolutional layer. For this reason, this padding mode has rarely been used in the literature, however, its merits and applicability are worth investigating.

These alternatives, while potentially beneficial for certain tasks [1, 12], can incur undesirable visual artifacts in the input of each layer, e.g. by making neighbors out of unrelated pixels, or can aggravate the problems of zero padding [1].

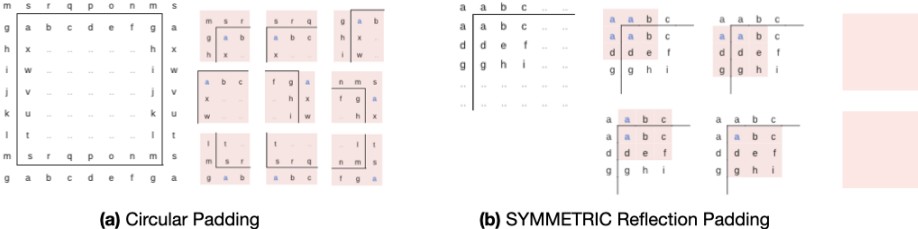

(a) Circular Padding      (b) SYMMETRIC Reflection Padding

Figure 6: `SAME` padding alternatives to 0-padding that warrant all pixels an equal number of kernel overlaps. Circular padding (a) utilizes all kernel cells uniformly as illustrated for one corner pixel, unlike `SYMMETRIC` padding (b) where the bottom and right cells do not overlap that pixel.

Another approach to deal with foveation is to couple CNNs with an attention mechanism akin to saccadic eye movements [11, 17, 18]. This can eliminate the need for padding and turn foveation into a useful property of CNNs. This property entails that the periphery of a CNN's input can be sampled at a lower resolution than the center without impacting accuracy, as shown by Deza et al [4, 5] on scene classification tasks. Such sampling of the retinal surface optimizes the visual acuity of human vision while limiting the number of receptors [7], in turn, reducing the metabolic cost. In fact a similar sampling was shown to emerge in an attention model with a learnable sampling grid [3], trained to attend in visual scenes through a translation-only glimpse window, akin to the human eye. This suggests that retinal sampling is the result of inherent foveation, not its source, and might offer an explanation for the emergence of center-periphery spatial organization in CNNs [14].

We hope our insights can help model developers be aware of potential foveation effects and their implications when designing their CNNs, and enable them to make informed choices that either mitigate or take advantage of these effects.

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

# 6 Appendix

We illustrate in detail how different padding algorithms treats the input pixels. For this purpose we illustrate step by step how each pixel is processed by the convolutional kernel. We choose a set of pixels that are sufficient to expose the behavior of the respective algorithm.

# VALID Padding   Illustrated on a 3x3 kernel

## Input

| a | b | c | .. | .. |
|---|---|---|----|----|
| d | e | f | .. | .. |
| g | h | i | .. | .. |
| .. | .. | .. | .. | .. |
| .. | .. | .. | .. | .. |

## # of kernel overlaps each pixel is involved in

| 1 | 2 | 3 | 3 | 3 |
|---|---|---|---|---|
| 2 | 4 | 6 | 6 | 6 |
| 3 | 6 | **9** | 9 | 9 |
| 3 | 6 | 9 | 9 | 9 |
| 3 | 6 | 9 | 9 | 9 |

## Which kernel cells these overlaps utilize?

a: 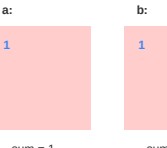 sum = 1

b: 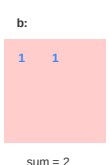 sum = 2

c: 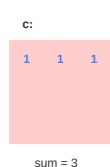 sum = 3

d: 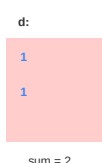 sum = 2

e: 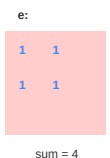 sum = 4

f: 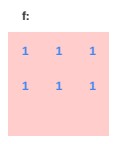 sum = 6

g: 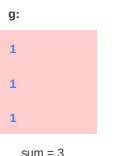 sum = 3

h: 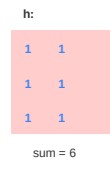 sum = 6

i: 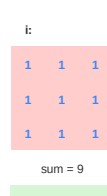 sum = 9

uniform

## Detailed Illustration of how the counts are derived

Overlaps involving (a)

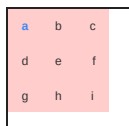

Overlaps involving (b)

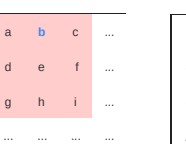

Overlaps involving (c)

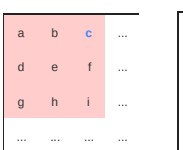 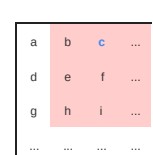 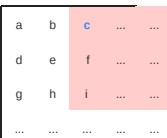

Overlaps involving (d): rotated version of (b)

Overlaps involving (e)

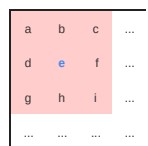 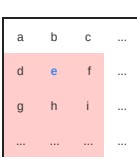 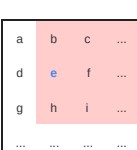 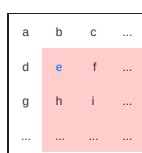

Overlaps involving (f)

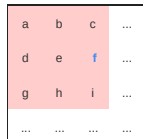 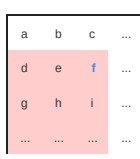 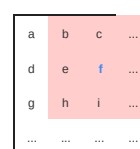 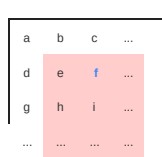 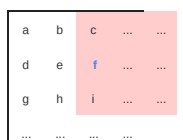 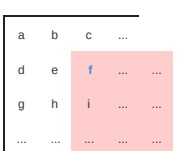

Overlaps involving (g): Rotated version of (c)

Overlaps involving (h): Rotated version of (f)

Overlaps involving (i): Regular uniform treatment

# Zero Padding

## Illustrated on 3x3 kernel and 1-pixel padding

**Original Input**

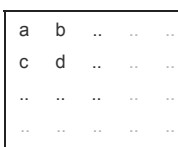

**Padded Input**

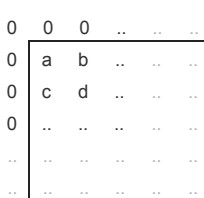

**# of kernel overlaps each pixel is involved in**

| 4 | 6 | 6 | 6 | 6 |
|---|---|---|---|---|
| 6 | 9 | 9 | 9 | 9 |
| 6 | 9 | 9 | 9 | 9 |
| 6 | 9 | 9 | 9 | 9 |
| 6 | 9 | 9 | 9 | 9 |

## Which kernel cells these overlaps utilize?

a 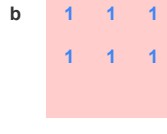    b 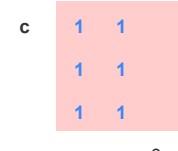    c 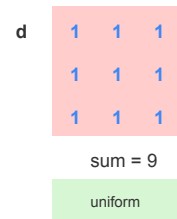    d

sum = 4          sum = 6          sum = 6          sum = 9

uniform

## Detailed Illustration of how the counts are derived

Overlaps involving (a)

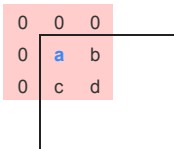    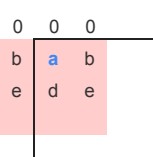    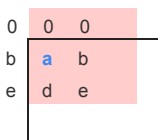

Overlaps involving (b)

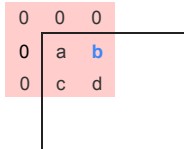    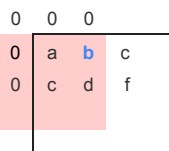    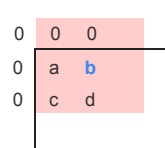    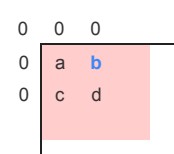    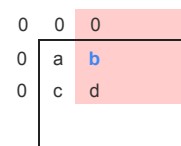    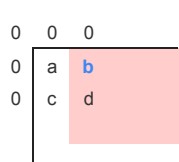

Other border cases are translation or rotation of (a) or (b)

# Circular Padding

## Illustrated on 3x3 kernel and 1-pixel padding

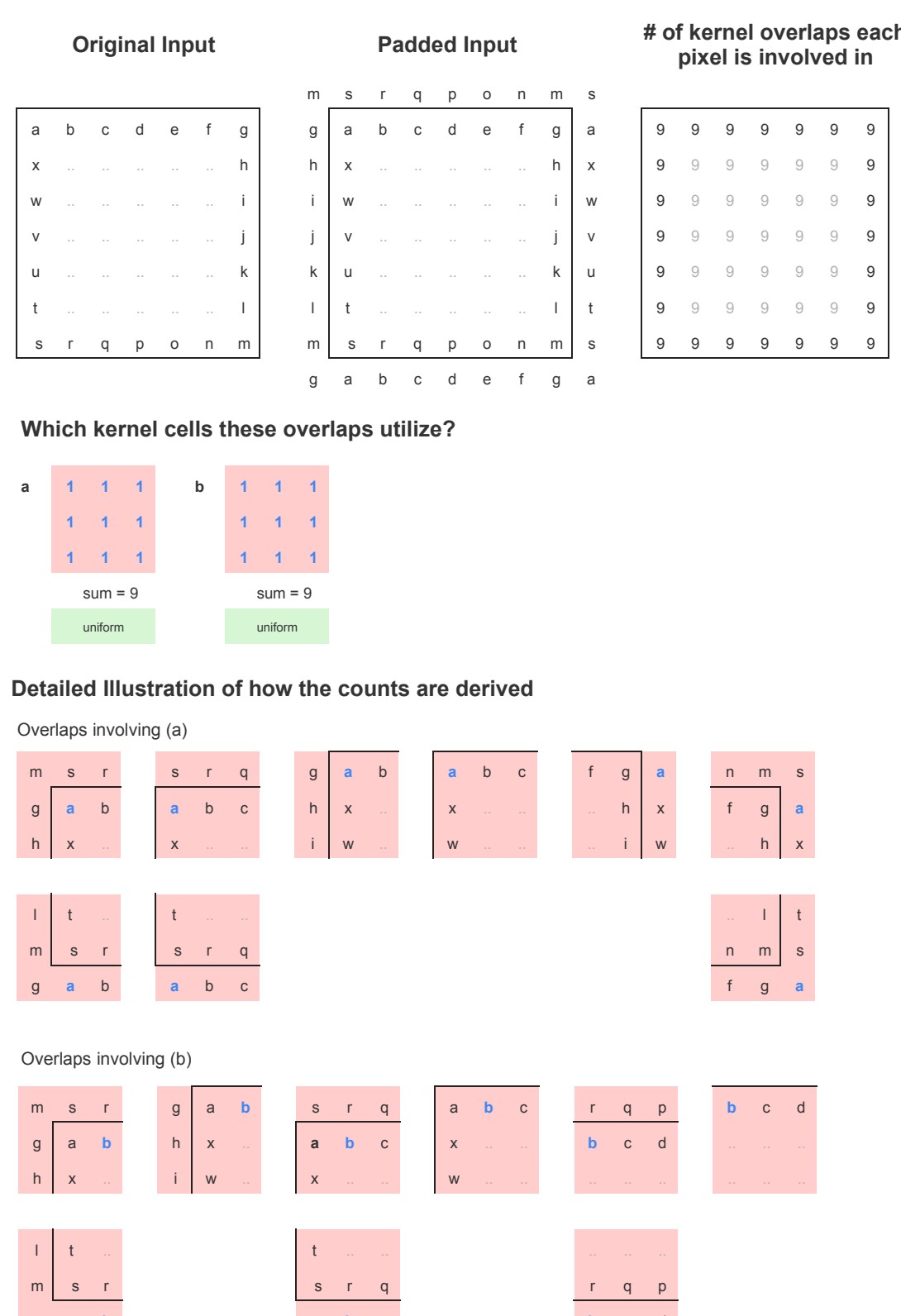

**Original Input**

**Padded Input**

**# of kernel overlaps each pixel is involved in**

**Which kernel cells these overlaps utilize?**

sum = 9    sum = 9

uniform    uniform

**Detailed Illustration of how the counts are derived**

Overlaps involving (a)

Overlaps involving (b)

Other border cases are translation or rotation of (a) or (b)

# Mirror Padding (SYMMETRIC)

## Illustrated on 3x3 kernel and 1-pixel padding

**Original Input**

| a | b | c | .. | .. |
|---|---|---|----|----|
| d | e | f | .. | .. |
| g | h | i | .. | .. |
| .. | .. | .. | .. | .. |
| .. | .. | .. | .. | .. |

**Padded Input**

|   | a | a | b | c | .. | .. |
|---|---|---|---|---|----|----|
| a | a | b | c | .. | .. | |
| d | d | e | f | .. | .. | |
| g | g | h | i | .. | .. | |
| .. | .. | .. | .. | .. | .. | |
| .. | .. | .. | .. | .. | .. | |

**# of kernel overlaps each pixel is involved in**

| 9 | 9 | 9 | 9 | 9 |
|---|---|---|---|---|
| 9 | 9 | 9 | 9 | 9 |
| 9 | 9 | 9 | 9 | 9 |
| 9 | 9 | 9 | 9 | 9 |
| 9 | 9 | 9 | 9 | 9 |

## Which kernel cells these overlaps utilize?

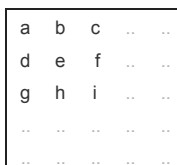

a:
| 4 | 2 |
|---|---|
| 2 | 1 |

sum = 9

b:
| 2 | 2 | 2 |
|---|---|---|
| 1 | 1 | 1 |

sum = 9

c:
| 2 | 2 | 2 |
|---|---|---|
| 1 | 1 | 1 |

sum = 9

d:
| 2 | 1 |
|---|---|
| 2 | 1 |
| 2 | 1 |

sum = 9

e:
| 1 | 1 | 1 |
|---|---|---|
| 1 | 1 | 1 |
| 1 | 1 | 1 |

sum = 9 · uniform

f:
| 1 | 1 | 1 |
|---|---|---|
| 1 | 1 | 1 |
| 1 | 1 | 1 |

sum = 9 · uniform

g:
| 2 | 1 |
|---|---|
| 2 | 1 |
| 2 | 1 |

sum = 9

h:
| 1 | 1 | 1 |
|---|---|---|
| 1 | 1 | 1 |
| 1 | 1 | 1 |

sum = 9 · uniform

i:
| 1 | 1 | 1 |
|---|---|---|
| 1 | 1 | 1 |
| 1 | 1 | 1 |

sum = 9 · uniform

## Detailed Illustration of how the counts are derived

Overlaps involving (a)

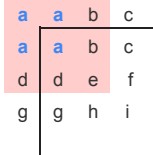

Overlaps involving (b)

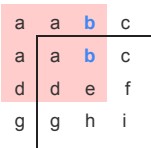 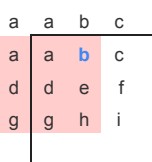 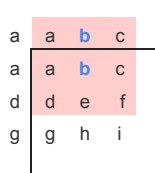 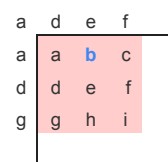 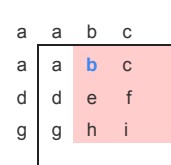

Overlaps involving (c)

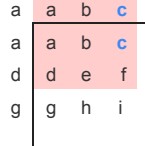 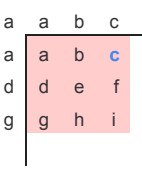 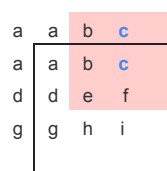

Overlaps involving (d): Rotated version of (b)

Overlaps involving (e)

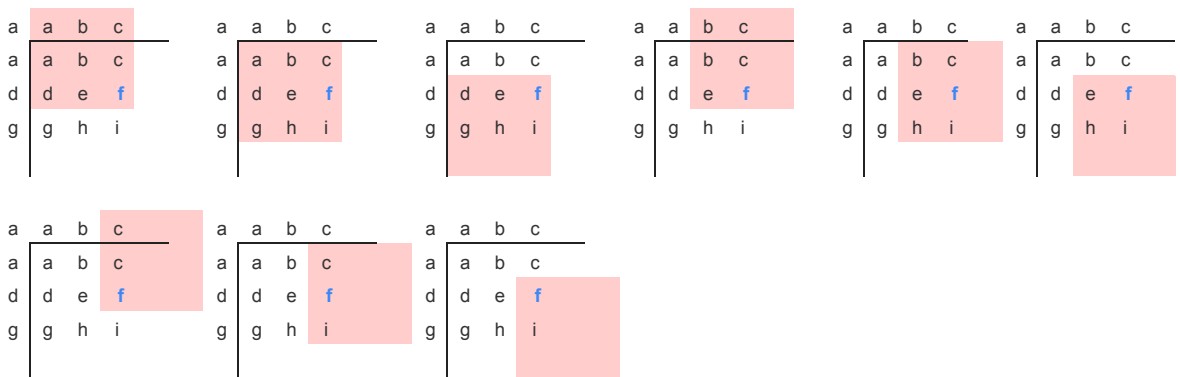

Overlaps involving (f)

Overlaps involving (g): Rotated version of (c)

Overlaps involving (h): Rotated version of (f)

Overlaps involving (i): Regular uniform treatment

# Replication Padding

## Illustrated on 5x5 kernel and 2-pixel padding

**Original Input**

**Padded Input**

**# of kernel overlaps for each pixel**

```
a  b  c  ..  ..
d  e  f  ..  ..
g  h  i  ..  ..
..  ..  ..  ..  ..
..  ..  ..  ..  ..
```

```
a  a  a  b  c  ..  ..
a  a  a  b  c  ..  ..
a  a  a  b  c  ..  ..
d  d  d  e  f  ..  ..
g  g  g  h  i  ..  ..
..  ..  ..  ..  ..
..  ..  ..  ..  ..
```

```
36  24  30  30  30
24  16  20  20  20
30  20  25  25  25
30  20  25  25  25
30  20  25  25  25
```

## Which kernel cells these overlaps utilize?

a:
```
9  6  3
6  4  2
3  2  1
```
sum = 36

b:
```
3  3  3
2  2  2
1  1  1
```
sum = 24

c:
```
3  3  3  3
2  2  2  2
1  1  1  1
```
sum = 30

d:
```
3  2  1
3  2  1
3  2  1
3  2  1
```
sum = 24

e:
```
1  1  1  1
1  1  1  1
1  1  1  1
1  1  1  1
```
sum = 16

f:
```
1  1  1  1  1
1  1  1  1  1
1  1  1  1  1
1  1  1  1  1
```
sum = 20

g:
```
3  2  1
3  2  1
3  2  1
3  2  1
3  2  1
```
sum = 30

h:
```
1  1  1  1
1  1  1  1
1  1  1  1
1  1  1  1
1  1  1  1
```
sum = 20

i:
```
1  1  1  1  1
1  1  1  1  1
1  1  1  1  1
1  1  1  1  1
1  1  1  1  1
```
sum = 25
uniform

## Detailed Illustration of how the counts are derived

### Convolutions involving (a)

### Convolutions involving (b)

### Convolutions involving (c)

Overlaps involving (d): Rotated version of (b)

Overlaps involving (e)

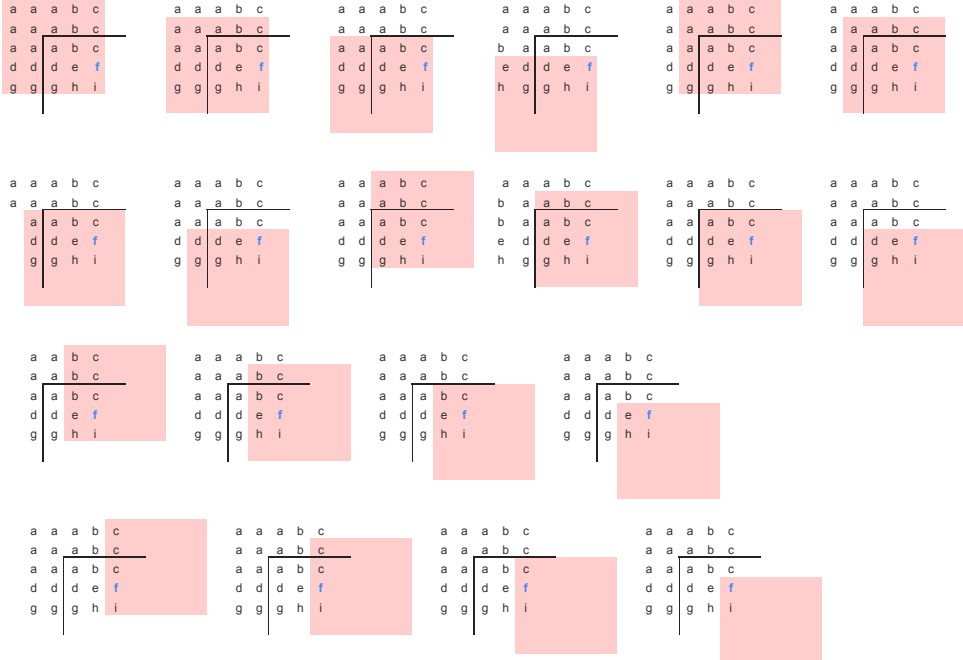

Overlaps involving (f)

Convolutions involving (g): Rotated version of (c)

Convolutions involving (h): Rotated version of (f)

Convolutions involving (i): Regular uniform treatment

# Mirror Padding (REFLECT)

## Illustrated on 3x3 kernel and 1-pixel padding

| Original Input | Padded Input | # of kernel overlaps each pixel is involved in |
|---|---|---|

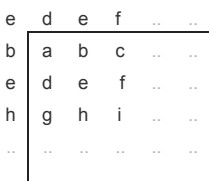

**Which kernel cells these overlaps utilize?**

| a: | b: | c: | d: | e: | f: | g: | h: | i: |
|---|---|---|---|---|---|---|---|---|

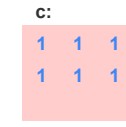

sum = 4 sum = 8 sum = 6 sum = 8 sum = 12 sum = 16 sum = 6 sum = 12 sum = 9

uniform

## Detailed Illustration of how the counts are derived

Overlaps involving (a)

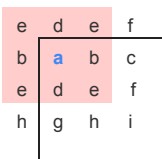 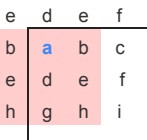 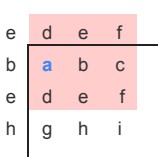 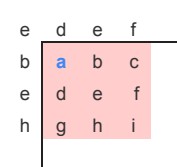

Overlaps involving (b)

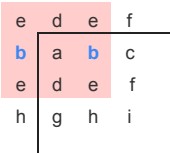 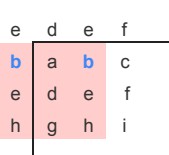 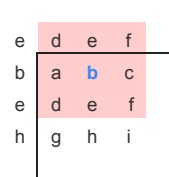 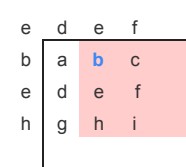

Overlaps involving (c)

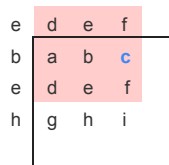 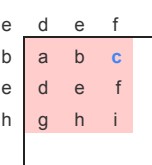 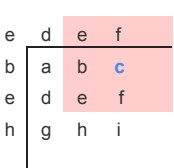 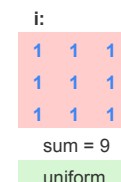

Overlaps involving (d): Rotated version of (b)

Overlaps involving (e)

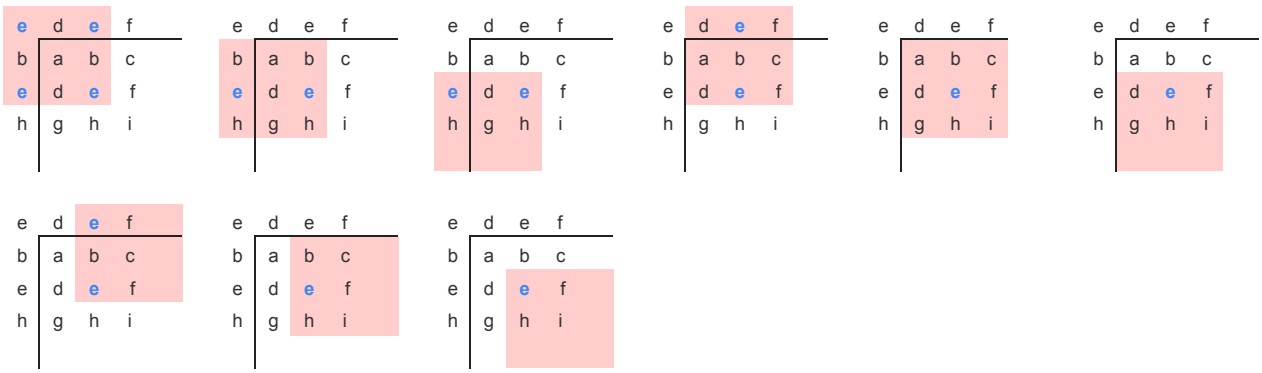

Overlaps involving (f)

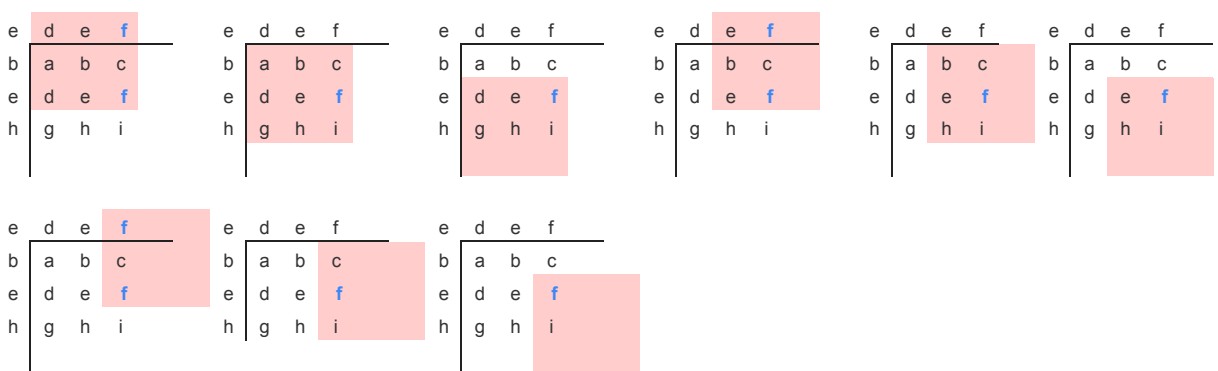

Overlaps involving (g): Rotated version of (c)

Overlaps involving (h): Rotated version of (f)

Overlaps involving (i): Regular uniform treatment