# OpenReview forum: "Are Convolutional Networks Inherently Foveated?"
_NeurIPS.cc/2021/Workshop/SVRHM — SVRHM 2021 Poster_

### Official Review · Reviewer_NceX · 2021-10-21
**Interesting analysis**

**Rating:** 6
**Confidence:** 4

**Review:**

This paper analyses the non-uniform effects of image pixels on the outputs of convolutional networks. The basic analysis is to count the number of output pixels that each input pixel influences. An equation is also given for the distance from the image border to the central region in which these counts are maximal.

These effects are qualitatively widely understood, but I haven’t seen them analyzed in detail as this paper does. The work is specifically relevant to the design of deep networks, particularly selection of a padding strategy, and it seems important to appreciate these effects generally. The analysis is thorough. Some parts of the paper are very clear, e.g. many example calculations are given in the appendix. However the figure captions could be more helpful, some terms could be defined more clearly, and it would help to give a derivation of Eq. 1.

Pros:
-	The analysis is useful, interesting, and (as far as I know) novel.
-	The result of Figure 4c is surprisingly large given what seems like a subtle change.

Cons:
-	I don’t think “foveated” is the right word. Biologically, the phenomenon addressed here may be more closely related to the tendency of deep visual neurons in inferotemporal cortex to have receptive fields around the middle of the visual field. Another somewhat less closely related biological phenomenon might be cortical magnification. I don’t think the biological fovea is related. The fovea is a region of peak visual acuity due to cell density in the retina, and one might stretch the idea to incorporate related phenomena (e.g. low optical aberrations, dominance of cones). But it’s independent of the phenomenon studied here, which is due to padding and stride within the network. I appreciate that the authors define their use of the term (line 38) but the definition isn’t consistent with a large existing literature.
-	I had trouble following some parts of the paper. For example, I had to study Figure 2b for some time before I knew what it meant (more caption would help here, e.g. a clear statement of the meaning of the shaded squares). Also Equation 1 is stated flatly without explanation. It’s less obvious than the counts of paths to output, which are given nine pages of examples in the appendix.

Minor points:
-	Personally, I found the term “spread of foveation” counter-intuitive. I thought it meant how much the fovea spread out over the image, rather than how far its border was from the edge.
-	In Figure 3b and d, the spread is the same, but d is non-uniform within the box in b. Please explain in the caption.
-	Pg 5: “Compared with zero padding, the overall pixel involvement is drastically reduced. This explains the strong impact of accuracy observed in [9].” Please explain this explanation.
-	Pg 7: “# of conv ops each pixel is involved in” This phrasing is strange because a single conv op spans the whole map.
-	Pg 7: “Which kernels cells these ops utilize” These are input pixels rather than ops.
-	Pg 7: Under the top left pink square the sum is 6.25. What is that the sum of?

---

### Official Review · Reviewer_tBDN · 2021-10-26
**An interesting remark on how architecture influences peripheral processing in CNNs**

**Rating:** 7
**Confidence:** 4

**Review:**

The authors show foveation-like effects in CNNs, due to border effects. As they describe, pixels at the border of an image fall in fewer receptive fields of the next layer than pixels in the center of the image. In this sense, peripheral pixels are “less processed” than central pixels. The authors call this effect “foveation”. Because this effect is cumulative, the “foveation” increases with network depth. The authors show that standardly used 0-padding mitigates but does not remove the effect. They provide 3 other methods for padding that successfully alleviate this effect.
While I don’t see this as a major advance for machine learning or neuroscience, it is still a worthwhile contribution in my opinion, and, as the authors explain, may help researchers make informed choices for their network architectures. Below, I list some comments and suggestions to improve the paper.

Majors:
-	In my opinion, calling this effect “foveation” is a little overblown. Initially, it reads as though the paper will show that CNNs have properties resembling human/primate/biological foveation. But this is not the case. Biological foveation does not really resemble the process described here, in which peripheral information is processed by fewer channels. Instead, biological foveation has many features like changes in RF size, cortical magnification, etc., which are not at all addressed here.
-	p.3: It is not clear exactly what “spread of foveation” means. As far as I understand from the math, it refers to how many pixels, taken from the edge of the image, are not part of the max number of kernels in the last layer. If this is true, then I find the name confusing, because it sounds as though it denotes the size of the foveal region, but in fact it quantifies the size of the NON-foveal region.

Minors:
-	The terms “inverted pyramid” and “inverted sub-pyramid” are employed without any explanation. Clarity would be improved by explaining them.
-	In the appendix about valid padding, the 3rd panel of “Convolutions involving (e)” is wrong: the red square should be shifted right.

---

### Official Review · Reviewer_S9T9 · 2021-10-28
**A comprehensive and insightful analysis of inherent foveation in CNNs**

**Rating:** 8
**Confidence:** 5

**Review:**

This paper analyses the contribution of input pixels to the output of a CNN in the context of the various hyperparameters and padding flavours to show that CNNs (at least those typically used in modern network architectures) are inherently foveated. The paper is well written and easy to follow. The definitions and approach contributed by this work serve to simplify and assist in reasoning about foveation in CNNs and this is a clear accept. In order to take this work further, it would be interesting to delve deeper into the consequences of this inherent foveation. Does removing or promoting inherent foveation impact performance? It would also be valuable to further explore the closing point made regarding the extent to which emergent foveation may be a consequence of the convolutional architecture. Overall, I believe that future versions of this work would benefit from streamlining the discussion of the impact of the various CNN options on inherent foveation in order to make room for a comprehensive re-assessment of prior art following the findings here.

---

### Official Review · Reviewer_o6Qk · 2021-11-01
**intriguing observations; the significance is unclear**

**Rating:** 5
**Confidence:** 4

**Review:**

The authors reported that when convolution networks using no paddling, the central pixels will have more ways to participate in the convolution (essentially a boundary problem). This leads to an implicit foveation of the input images, which persist even when padding is used as the authors argued. The authors quantifies the foveation behavior under a few different CNN setups, focusing on the spread and the magnitude.

This submission contains some interesting and intriguing observations. However, the implications of these observations regarding developing better algorithms remain unclear.  The authors claims their findings in CNN are of “the same nature and have similar implications” as the foveation in human. I found this assertion to be largely unconvincing.

---

### Author Response · Authors · 2021-12-08
**Revision Summary**

We thank all our reviewers for the very helpful comments. These comments have triggered several updates to the manuscript and further experimentation to study the implications of the effects we report about.

Three of our reviewers have valid concerns around the similarity between these effects and foveation in human vision. We rephrased the title to a question to avoid the unwarranted impression that we have definitive answers about the similarity. We cited three references that also draw a similar connection:
- Cheung et al (ICLR 2016): [Emergence of **foveal image sampling** from learning to attend in visual scenes](https://openreview.net/forum?id=SJJKxrsgl).
- Luo et al. (NeurIPS 2016). [Understanding the effective receptive field in deep convolutional neural networks](https://proceedings.neurips.cc/paper/2016/file/c8067ad1937f728f51288b3eb986afaa-Paper.pdf). (p.8 "our analysis [...] suggests that convolutional networks may automatically create a form of **foveal representation**.").
- Mohsenzadeh et al. (Scientific reports 2020): [Emergence of visual **center-periphery spatial organization** in deep convolutional neural networks](https://www.nature.com/articles/s41598-020-61409-0).

Our use of the word foveation is hence mainly to draw a provocative and rough analogy. We do think that the foveal sampling in human vision could have possibly emerged as a result of early vision mechanisms: If these mechanisms inherently put less weight on the periphery compared with the center, there is no benefit of retaining a uniform sampling across both regions. Instead, focusing visual acuity in the center would make best use of the visual processing mechanisms. In other words, this sampling (and the subsequent cortical magnification) could be the result of inherently foveated visual processing, not its source.

We also rephrased the term "spread of foveation" into "extent of non-foveal area", and defined what we mean by "inverted sub-pyramids" as Reviewer tBDN suggests. Finally, we improved Figure 2 and Figure 3, explained how our results explain the observations in [9], and updated the illustrations on page 7 onwards as Reviewer NceX suggests.

We hope our contribution will be a valuable addition to the workshop, and are eager to discuss our findings with the SVRHM community at NeurIPS 2021.

---

### Decision · Program_Chairs · 2021-11-02

Accept (Poster)